# Learning bilingual word embeddings with (almost) no bilingual data

## Abstract

Most methods to learn bilingual word embeddings rely on large parallel corpora, which is difficult to obtain for most language pairs. This has motivated an active research line to relax this requirement, with methods that use document-aligned corpora or bilingual dictionaries of a few thousand words instead. In this work, we further reduce the need of bilingual resources using a very simple self-learning approach that can be combined with any dictionary-based mapping technique. Our method exploits the structural similarity of embedding spaces, and works with as little bilingual evidence as a 25 word dictionary or even an automatically generated list of numerals, obtaining results comparable to those of systems that use richer resources.

## 1 Introduction

Multilingual word embeddings have attracted a lot of attention in recent times. In addition to having a direct application in inherently cross-lingual tasks like machine translation (Zou et al., 2013) and cross-lingual entity linking (Tsai and Roth, 2016), they provide an excellent mechanism for transfer learning, where a model trained in a resource-rich language is transferred to a less-resourced one, as shown with part-of-speech tagging (Zhang et al., 2016), parsing (Xiao and Guo, 2014) and document classification (Klementiev et al., 2012), among others.

Most methods to learn these multilingual word embeddings make use of large parallel corpora (Gouws et al., 2015; Luong et al., 2015), but there have been several proposals to relax this requirement, given its scarcity in most language pairs. A possible relaxation is to use document-aligned or label-aligned comparable corpora (Søgaard et al., 2015; Vulic and Moens, 2016; Mogadala and Rettinger, 2016), but large amounts of such corpora are not always available for some language pairs.

An alternative approach that we follow here is to independently train the embeddings for each language on monolingual corpora, and then learn a linear transformation to map the embeddings from one space into the other by minimizing the distances in a bilingual dictionary, usually in the range of a few thousand entries (Mikolov et al., 2013a; Artetxe et al., 2016). However, dictionaries of that size are not readily available for many language pairs, specially those involving less-resourced languages.

In this work, we reduce the need of large bilingual dictionaries to much smaller seed dictionaries. Our method can work with as little as 25 word pairs, which are straightforward to obtain assuming some basic knowledge of the languages involved. The method can also work with trivially generated seed dictionaries of numerals (i.e. 1-1, 2-2, 3-3, 4-4...) making it possible to learn bilingual word embeddings without any real bilingual data. In either case, we obtain very competitive results, comparable to other state-of-the-art methods that make use of much richer bilingual resources.

The proposed method is an extension of existing mapping techniques, where the dictionary is used to learn the embedding mapping and the embedding mapping is used to induce a new dictionary iteratively in a self-learning fashion (see Figure 1). In spite of its simplicity, our analysis of the implicit optimization objective reveals that the method is exploiting the structural similarity of independently trained embeddings.

Section 2 analyzes previous work. Section 3 describes the self-learning framework, while Section 4 presents the experiments. Section 5 analyzes the underlying optimization objective.

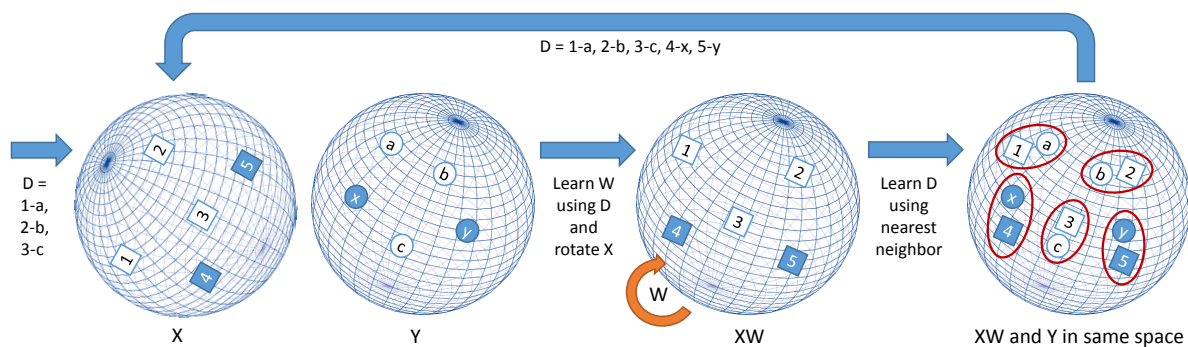

Figure 1: A general schema of the proposed self-learning framework. Previous works learn a mapping W based on the seed dictionary D, which is then used to learn the full dictionary. In our proposal we use the new dictionary to learn a new mapping, iterating until convergence.

## 2 Related work

We will first focus on bilingual embedding mappings, which are the basis of our proposals, and then on other unsupervised and weakly supervised methods to learn bilingual word embeddings.

### 2.1 Bilingual embedding mappings

Methods to induce bilingual mappings work by independently learning the embeddings in each language using monolingual corpora, and then learning a transformation from one embedding space into the other based on a bilingual dictionary.

The first of such methods is due to Mikolov et al. (2013a), who learn the linear transformation that minimizes the sum of squared Euclidean distances for the dictionary entries. The same optimization objective is used by Zhang et al. (2016), who constrain the transformation matrix to be orthogonal. Xing et al. (2015) incorporate length normalization in the training of word embeddings and maximize the cosine similarity instead, enforcing the orthogonality constraint to preserve the length normalization after the mapping. Finally, Lazaridou et al. (2015) use max-margin optimization with intruder negative sampling.

While all the previous methods learn a single linear transformation from the source language into the target language, Faruqui and Dyer (2014) use canonical correlation analysis to learn a separate linear transformation for each language that maps them to a shared vector space. Lu et al. (2015) extend this work and apply deep canonical correlation analysis to learn non-linear transformations.

Artetxe et al. (2016) propose a general framework that clarifies the relation between Mikolov et al. (2013a), Xing et al. (2015), Faruqui and Dyer (2014) and Zhang et al. (2016) as variants of the same core optimization objective, and show that a new variant is able to surpass them all. While most of the previous methods use gradient descent, Artetxe et al. (2016) propose an efficient analytical implementation for those same methods, which we use in this work.

A prominent application of bilingual embedding mappings, with a direct application in machine translation (Zhao et al., 2015), is bilingual lexicon extraction, which is also the main evaluation method for bilingual embedding mappings. More specifically, the learned mapping is used to induce the translation of source language words that were missing in the original dictionary, usually by taking their nearest neighbor word in the target language according to cosine similarity, although Dinu et al. (2015) propose a more complex globally-corrected approach.

### 2.2 Unsupervised and weakly supervised bilingual embeddings

As mentioned before, our method works with as little as 25 word pairs, while the methods discussed previously use thousands of pairs. The only exception in this regard is the work by Zhang et al. (2016), who only use 10 word pairs to learn the orthogonal mapping minimizing the sum of squared Euclidean distances, with good results on transfer learning for part-of-speech tagging. Our experiments will show that, although their method captures coarse-grained relations, it fails on finer-grained tasks like bilingual lexicon induction.

Miceli Barone (2016) attempts to learn bilingual embedding mappings without any bilingual

---

**Algorithm 1** Traditional framework

**Input:** $X$ (source embeddings)
**Input:** $Z$ (target embeddings)
**Input:** $D$ (seed dictionary)
1: $W \leftarrow$ LEARN_MAPPING$(X, Z, D)$
2: $D \leftarrow$ LEARN_DICTIONARY$(X, Z, W)$
3: EVALUATE_DICTIONARY$(D)$

---

data. He uses adversarial autoencoders (Makhzani et al., 2016), combining an encoder that maps the source language embeddings into the target language, a decoder that reconstructs the original embeddings, and a discriminator that distinguishes mapped embeddings from real target language embeddings. Although promising, the reported performance of the learned embeddings is poor in comparison to other methods.

Finally, the induction of bilingual knowledge from monolingual corpora is closely related to the decipherment scenario, for which models that incorporate word embeddings have also been proposed (Dou et al., 2015). However, decipherment is only concerned with translating text from one language to another and relies on complex statistical models that are designed specifically for that purpose, while our approach is more general and learns task-independent multilingual embeddings.

## 3 Proposed self-learning framework

As discussed in Section 2.1, a common evaluation task (and practical application) of bilingual embedding mappings is to induce bilingual lexicons, that is, to obtain the translation of source words that were missing in the training dictionary, which are then compared to a gold standard test dictionary for evaluation. This way, one can say that the seed (train) dictionary is used to learn a mapping, which is then used to induce a better dictionary (at least in the sense that it is larger). Algorithm 1 summarizes this framework.

Following this observation, we propose to use the output dictionary in Algorithm 1 as the input of the same system in a self-learning fashion which, assuming that the output dictionary was indeed better than the original one, should serve to learn a better mapping and, consequently, an even better dictionary the second time. The process can then be repeated iteratively to obtain a hopefully better mapping and dictionary each time until some convergence criterion is met. Algorithm 2 summa-

---

**Algorithm 2** Proposed self-learning framework

**Input:** $X$ (source embeddings)
**Input:** $Z$ (target embeddings)
**Input:** $D$ (seed dictionary)
1: **repeat**
2:    $W \leftarrow$ LEARN_MAPPING$(X, Z, D)$
3:    $D \leftarrow$ LEARN_DICTIONARY$(X, Z, W)$
4: **until** convergence criterion
5: EVALUATE_DICTIONARY$(D)$

---

rizes this alternative framework that we propose.

Our method can be combined with any embedding mapping and dictionary induction technique (see Section 2.1). However, efficiency turns out to be critical for a variety of reasons. First of all, by enclosing the learning logic in a loop, the total training time is increased by the number of iterations. Even more importantly, our framework requires to explicitly build the entire dictionary at each iteration, whereas previous work tends to induce the translation of individual words on-demand later at runtime. Moreover, from the second iteration onwards, it is this induced, full dictionary that has to be used to learn the embedding mapping, and not the considerably smaller seed dictionary as it is typically done. In the following two subsections, we respectively describe the embedding mapping method and the dictionary induction method that we adopt in our work with these efficiency requirements in mind.

### 3.1 Embedding mapping

As discussed in Section 2.1, most previous methods to learn embedding mappings use variants of gradient descent. As the only exceptions in this regard, both Faruqui and Dyer (2014) and Artetxe et al. (2016) provide more efficient exact methods, with the second being simpler, more efficient, and yielding better results as reported in their paper. In the following, we present their method, explicitly incorporating the dictionary in the formalization so it can be smoothly integrated in our self-learning algorithm.

Let $X$ and $Z$ denote the word embedding matrices in two languages so that $X_{i*}$ corresponds to the $i$th source language word embedding and $Z_{j*}$ corresponds to the $j$th target language embedding. While Artetxe et al. (2016) assume these two matrices are aligned according to the dictionary, we drop this assumption and represent the dictionary explicitly as a binary matrix $D$, so that $D_{ij} = 1$

if the $i$th source language word is aligned with the $j$th target language word. The goal is then to find the optimal mapping matrix $W^*$ so that the sum of squared Euclidean distances between the mapped source embeddings $X_{i*}W$ and target embeddings $Z_{j*}$ for the dictionary entries $D_{ij}$ is minimized:

$$W^* = \arg\min_W \sum_i \sum_j D_{ij} ||X_{i*}W - Z_{j*}||^2$$

Following Artetxe et al. (2016), we length normalize and mean center the embedding matrices $X$ and $Z$ in a preprocessing step, and constrain $W$ to be an orthogonal matrix (i.e. $WW^T = W^TW = I$), which serves to enforce monolingual invariance, preventing a degradation in monolingual performance while yielding to better bilingual mappings. Under such orthogonality constraint, minimizing the squared Euclidean distance becomes equivalent to maximizing the dot product, so the above optimization objective can be reformulated as follows:

$$W^* = \arg\max_W \text{Tr}\left(XWZ^TD^T\right)$$

where $\text{Tr}(\cdot)$ denotes the trace operator (the sum of all the elements in the main diagonal). The optimal orthogonal solution for this problem is given by $W^* = UV^T$, where $X^TDZ = U\Sigma V^T$ is the singular value decomposition of $X^TDZ$. Since the dictionary matrix $D$ is sparse, this can be efficiently computed in linear time with respect to the number of dictionary entries.

### 3.2 Dictionary induction

As discussed in Section 2.1, practically all previous work uses nearest neighbor retrieval for word translation induction based on embedding mappings. In nearest neighbor retrieval, each source language word is assigned the closest word in the target language. In our work, we use the dot product between the mapped source language embeddings and the target language embeddings as the similarity measure, which is roughly equivalent to cosine similarity given that we apply length normalization followed by mean centering as a preprocessing step (see Section 3.1). This way, following the notation in Section 3.1, we set $D_{ij} = 1$ if $j = \text{argmax}_k (X_{i*}W) \cdot Z_{k*}$ and $D_{ij} = 0$ otherwise[1].

---

[1]Note that we induce the dictionary entries starting from the source language words. We experimented with other alternatives in development, with minor differences.

While we find that independently computing the similarity measure between all word pairs is prohibitively slow, the computation of the entire similarity matrix $XWZ^T$ can be easily vectorized using popular linear algebra libraries, obtaining big performance gains. However, the resulting similarity matrix is often too large to fit in memory when using large vocabularies. For that reason, instead of computing the entire similarity matrix $XWZ^T$ in a single step, we iteratively compute submatrices of it using vectorized matrix multiplication, find their corresponding minima each time, and then combine the results.

## 4 Experiments and results

In this section, we experimentally test the proposed method in bilingual lexicon induction and crosslingual word similarity. Subsection 4.1 describes the experimental settings, while Subsections 4.2 and 4.3 present the results obtained in each of the tasks. The code and resources necessary to reproduce our experiments are available at https://github.com/XXXXX[2].

### 4.1 Experimental settings

For easier comparison with related work, we evaluated our mappings on **bilingual lexicon induction** using the public **English-Italian** dataset by Dinu et al. (2015), which includes monolingual word embeddings in both languages together with a bilingual dictionary split in a training set and a test set[3]. The embeddings were trained with the word2vec toolkit with CBOW and negative sampling (Mikolov et al., 2013b)[4], using a 2.8 billion word corpus for English (ukWaC + Wikipedia + BNC) and a 1.6 billion word corpus for Italian (itWaC). The training and test sets were derived from a dictionary built form Europarl word alignments and available at OPUS (Tiedemann, 2012), taking at random 1,500 uniformly distributed entries as the test set and the 5,000 most frequent of the remaining word pairs as the training set.

In addition to English-Italian, we selected two other languages from different language families with publicly available resources. We thus cre-

---

[2]To be released upon acceptance.
[3]http://clic.cimec.unitn.it/~georgiana.dinu/down/
[4]The context window was set to 5 words, the dimension of the embeddings to 300, the sub-sampling to 1e-05 and the number of negative samples to 10, and the vocabulary was restricted to the 200,000 most frequent words

| | English-Italian | | | English-German | | | English-Finnish | | |
| --- | --- | --- | --- | --- | --- | --- | --- | --- | --- |
| | 5,000 | 25 | num. | 5,000 | 25 | num. | 5,000 | 25 | num. |
| Mikolov et al. (2013a) | 34.93% | 0.00% | 0.00% | 35.00% | 0.00% | 0.07% | 25.91% | 0.00% | 0.00% |
| Xing et al. (2015) | 36.87% | 0.00% | 0.13% | 41.27% | 0.07% | 0.53% | 28.23% | 0.07% | 0.56% |
| Zhang et al. (2016) | 36.73% | 0.07% | 0.27% | 40.80% | 0.13% | 0.87% | 28.16% | 0.14% | 0.42% |
| Artetxe et al. (2016) | 39.27% | 0.07% | 0.40% | **41.87%** | 0.13% | 0.73% | **30.62%** | 0.21% | 0.77% |
| Our method | **39.67%** | **37.27%** | **39.40%** | 40.87% | **39.60%** | **40.27%** | 28.72% | **28.16%** | **26.47%** |

Table 1: Accuracy on bilingual lexicon induction for different seed dictionaries

ated analogous datasets for **English-German** and **English-Finnish**. In the case of German, the embeddings were trained on the 0.9 billion word corpus SdeWaC, which is part of the WaCky collection (Baroni et al., 2009), which was also used for English and Italian. Given that Finnish is not included in this collection, we used the 2.8 billion word Common Crawl corpus provided at WMT 2016[5] instead, which we tokenized using the Stanford Tokenizer (Manning et al., 2014). In addition to that, we created training and test sets for both pairs from their respective Europarl dictionaries from OPUS following the exact same procedure used for English-Italian, and the word embeddings were also trained using the same configuration as Dinu et al. (2015).

Given that the main focus of our work is on **small seed dictionaries**, we created random subsets of 2,500, 1,000, 500, 250, 100, 75, 50 and 25 entries from the original training dictionaries of 5,000 entries. This was done by shuffling once the training dictionaries and taking their first $k$ entries, so it is guaranteed that each dictionary is a strict subset of the bigger dictionaries.

In addition to that, we explored using automatically generated dictionaries as a shortcut to practical unsupervised learning. For that purpose, we created **numeral dictionaries**, consisting of words matching the `[0-9]+` regular expression in both vocabularies (e.g. 1-1, 2-2, 3-3, 1992-1992 etc.). The resulting dictionary had 2772 entries for English-Italian, 2148 for English-German, and 2345 for English-Finnish. While more sophisticated approaches are possible (e.g. involving the edit distance of all words), we believe that this method is general enough that should work with practically any language pair, as arabic numerals are often used even in languages with a different writing system (e.g. Chinese and Russian).

While bilingual lexicon induction is a standard

evaluation task for seed dictionary based methods like ours, it is unsuitable for bilingual corpus based methods, as statistical word alignment already provides a reliable way to derive dictionaries from bilingual corpora and, in fact, this is how the test dictionary itself is built in our case. For that reason, we carried out some experiments in **crosslingual word similarity** as a way to test our method in a different task and allowing compare it to systems that use richer bilingual data. There are no many crosslingual word similarity datasets, and we used the RG-65 and WordSim-353 crosslingual datasets for English-German and the WordSim-353 crosslingual dataset for English-Italian as published by Camacho-Collados et al. (2015) [6].

As for the **convergence criterion** required by our method, we decide to stop training when the improvement on the average similarity for the automatically generated dictionary falls below a given threshold from one iteration to the next. As the average similarity ranges from -1 to 1, we decide to set this threshold at 1e-6, which we find to be a very conservative value yet enough that training takes a reasonable amount of time. The curves in the next section show that this threshold was a reasonable choice.

This convergence criterion is usually met in less than 100 iterations, each of them taking 5 minutes on a modest desktop computer (Intel Core i5-4670 CPU with 8GiB of RAM), including the induction of a dictionary of 200,000 words at each iteration.

### 4.2 Bilingual lexicon induction

For the experiments on bilingual lexicon induction, we compared our method with those proposed by Mikolov et al. (2013a), Xing et al. (2015), Zhang et al. (2016) and Artetxe et al. (2016)[7]. The results obtained with the 5,000 en-

---

[5] http://www.statmt.org/wmt16/ translation-task.html

[6] http://lcl.uniroma1.it/ similarity-datasets/

[7] We used the publicly available implementation at https://github.com/artetxem/vecmap

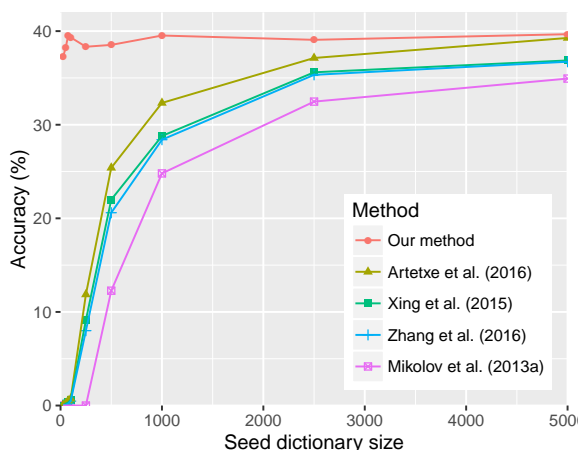

Figure 2: Accuracy on English-Italian bilingual lexicon induction for different seed dictionaries

try, 25 entry and the numerals dictionaries for all the 3 language pairs are given in Table 1.

The results for the 5,000 entry dictionaries show that our method is comparable or even better than the other systems. As another reference, the best published results for the full English-Italian dictionary are due to Lazaridou et al. (2015)[8], who report an accuracy of 40.20%, almost at pair with our system (39.67%).

In any case, the main focus of our work is on smaller dictionaries, and it is under this setting that our method really stands out. The 25 entry and numerals columns in Table 1 show the results for this setting, where all previous methods drop dramatically, falling below 1% accuracy in all cases. The method by Zhang et al. (2016) also obtains poor results with small dictionaries, which reinforces our hypothesis in Section 2.2 that their method can only capture coarse-grain bilingual relations for small dictionaries. In contrast, our proposed method obtains very competitive results for all dictionaries, with a difference of only 1-2 points between the full dictionary and both the 25 entry dictionary and the numerals dictionary in all three languages. Figure 2 shows the curve of the English-Italian accuracy for different seed dictionary sizes, confirming this trend.

Finally, it is worth mentioning that, even if all the three language pairs show the same general behavior, there are clear differences in their ab-

solute accuracy numbers, which can be attributed to the linguistic proximity of the languages involved. In particular, the results for English-Finnish are about 10 points below the rest, which are explained by the fact that Finnish is a non-indoeuropean agglutinative language, making the task considerably more difficult for this language pair. In this regard, we believe that the good results with small dictionaries are a strong indication of the robustness of our method, showing that it is able to learn good bilingual mappings from very little bilingual evidence even for distant language pairs where the structural similarity of the embedding spaces is presumably weaker.

### 4.3 Crosslingual word similarity

In addition to the baseline systems in Section 4.2, in the crosslingual similarity experiments we also tested the method by Luong et al. (2015), which is the state-of-the-art for bilingual word embeddings based on parallel corpora (Upadhyay et al., 2016)[9]. As this method is an extension of word2vec, we used the same hyperparameters as for the monolingual embeddings when possible (see Section 4.1), and leave the default ones otherwise. We used Europarl as our parallel corpus to train this method as done by the authors, which consists of nearly 2 million parallel sentences.

As shown in the results in Table 2, our method obtains the best results in all cases, surpassing the rest of the dictionary-based methods by 1-3 points depending on the dataset. But, most importantly, it does not suffer from any significant degradation for using smaller dictionaries and, in fact, our method gets better results using the 25 entry dictionary or the numeral list as the only bilingual evidence than any of the baseline systems using much richer resources.

The relatively poor results of Luong et al. (2015) can be attributed to the fact that the dictionary based methods make use of much bigger monolingual corpora, while methods based on parallel corpora are restricted to smaller corpora. However, it is not clear how to introduce monolingual corpora on those methods. We did run some experiments with BilBOWA (Gouws et al., 2015), which supports training in monolingual corpora in

---

[8]The code for this method is not publicly available but, given that our best baseline system obtains similar results on English-Italian (39.27% vs 40.20%), we consider that the state-of-the-art is already well represented in our experiments.

[9]We also tested English-German pre-trained embeddings from Klementiev et al. (2012) and Chandar A P et al. (2014). They both had coverage problems that made the results hard to compare, and, when considering the correlations for the word pairs in their vocabulary, their performance was poor.

|  |  | IT | DE | |
|---|---|---|---|---|
|  | Bi. data | WS | RG | WS |
| Luong et al. (2015) | Europarl | .331 | .335 | .356 |
| Mikolov et al. (2013a) | 5k dict | .627 | .643 | .528 |
| Xing et al. (2015) | 5k dict | .614 | .700 | .595 |
| Zhang et al. (2016) | 5k dict | .61.6 | .704 | .596 |
| Artetxe et al. (2016) | 5k dict | .617 | .716 | .597 |
| Our method | 5k dict | .624 | .742 | **.616** |
|  | 25 dict | .626 | **.749** | .612 |
|  | num. | **.628** | .739 | .604 |

Table 2: Spearman correlations on English-Italian and English-German crosslingual word similarity

addition to bilingual corpora, but obtained very poor results[10]. All in all, our experiments show that it is better to use large monolingual corpora in combination with very little bilingual data rather than a bilingual corpus of a standard size alone.

## 5 Global optimization objective

It might seem somehow surprising at first that, as seen in the previous section, our simple self-learning approach is able to learn high quality bilingual embeddings from small seed dictionaries instead of falling in degenerated solutions. In this section, we try to shed light on our approach, and give empirical evidence supporting our claim.

More concretely, we argue that, for the embedding mapping and dictionary induction methods described in Section 3, the proposed self-learning framework is implicitly solving the following global optimization problem[11]:

$$W^* = \arg\max_{W} \sum_{i} \max_{j} (X_{i*}W) \cdot Z_{j*}$$
$$\text{s.t.} \quad WW^T = W^TW = I$$

Contrary to the optimization objective for $W$ in Section 3.1, the global optimization objective does not refer to any dictionary, and maximizes the similarity between each source language word and its closest target language word. Intuitively, a random solution would map source language embeddings to seemingly random locations in the target language space, and it would thus be unlikely that they have any target language word nearby, making the optimization value small. In contrast, a good solution would map source language words

___
[10]Upadhyay et al. (2016) report similar problems using BilBOWA.

[11]While we restrict our formal analysis to the embedding mapping and dictionary induction method that we use, the general reasoning should be valid for other choices as well.

close to their translation equivalents in the target language space, and they would thus have their corresponding embeddings nearby, making the optimization value large. While it is certainly possible to build degenerated solutions that take high optimization values for small subsets of the vocabulary, we think that the structural similarity between independently trained embedding spaces in different languages is strong enough that optimizing this function yields to meaningful bilingual mappings when the size of the vocabulary is much larger than the dimensionality of the embeddings.

The reasoning for how the self-learning framework is optimizing this objective is as follows. At the end of each iteration, the dictionary $D$ is updated to assign, for the current mapping $W$, each source language word to its closest target language word. This way, when we update $W$ to maximize the average similarity of these dictionary entries at the beginning of the next iteration, it is guaranteed that the value of the optimization objective will improve (or at least remain the same). The reason is that the average similarity between each word and what were previously the closest word will be improved if possible, as this is what the updated $W$ directly optimizes (see Section 3.1). In addition to that, it is also possible that, for some source words, some other target words get closer after the update. Thanks to this, our self-learning algorithm is guaranteed to converge to a local optimum of the above global objective, behaving like an alternating optimization algorithm for it.

It is interesting to note that the above reasoning is valid no matter what the the initial solution is, and, in fact, the optimization objective does not depend on the seed dictionary nor any other bilingual resource. For that reason, it should be possible to use a random initialization instead of a small seed dictionary. However, we empirically observe that this works poorly in practice, as our algorithm tends to get stuck in poor local optima when the initial solution is not good enough.

The general behavior of our method is reflected in Figure 3, which shows the learning curve for different seed dictionaries according to both the objective function and the accuracy on bilingual lexicon induction. As it can be seen, the objective function is improved from iteration to iteration and converges to a local optimum just as expected. At the same time, the learning curves show a strong correlation between the optimization objective and

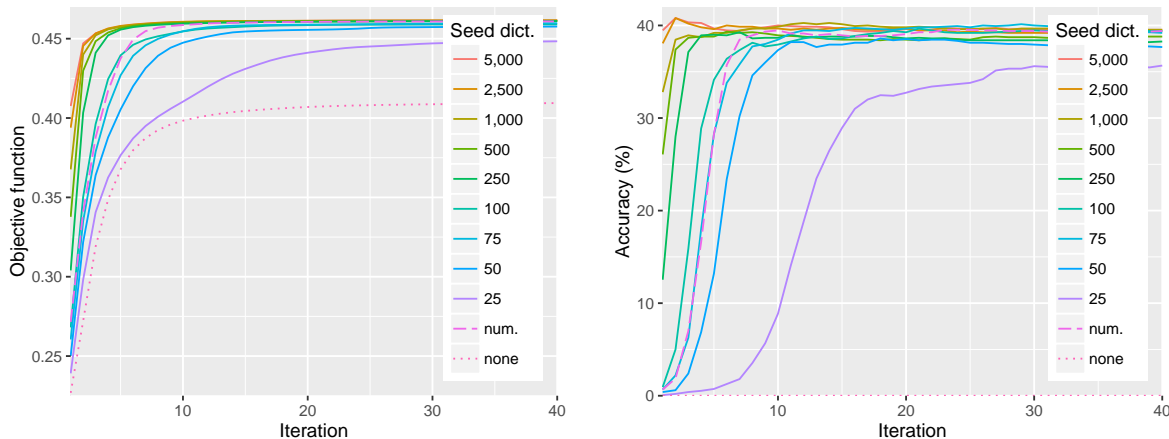

Figure 3: Learning curve on English-Italian according to the global objective function (left) and the accuracy on bilingual lexicon induction (right)

the accuracy, as it can be clearly observed that improving the former leads to an improvement of the latter, confirming our explanations. Regarding random initialization, the figure shows that the algorithm gets stuck in a poor local optimum of the objective function, which is the reason of the bad performance (0% accuracy) on bilingual lexicon induction, but the proposed optimization objective itself seems to be adequate.

Finally, we empirically observe that our algorithm learns similar mappings no matter what the seed dictionary was. We first repeated our experiments on English-Italian bilingual lexicon induction for 5 different dictionaries of 25 entries, obtaining an average accuracy of 38.15% and a standard deviation of only 0.75%. In addition to that, we observe that the overlap between the predictions made when starting with the full dictionary and the numerals dictionary is 76.00% (60.00% for the 25 entry dictionary). At the same time, 37.00% of the test cases are correctly solved by both instances, and it is only 5.07% of the test cases that one of them gets right and the other wrong (34.00% and 8.94% for the 25 entry dictionary). This clearly shows that these different systems do not only obtain a similar accuracy, but also have a very similar behavior. In other words, our algorithm tends to converge to similar solutions even for disjoint seed dictionaries, which is in line with our view that we are implicitly optimizing an objective that is independent from the seed dictionary, yet a seed dictionary is necessary to build a good enough initial solution to avoid getting stuck in poor local optima. For that reason, it is likely that better methods to tackle this opti-

mization problem would allow learning bilingual word embeddings without any bilingual evidence at all and, in this regard, we believe that our work opens exciting opportunities for future research.

## 6 Conclusions and future work

In this work, we propose a simple self learning framework to learn bilingual word embedding mappings in combination with any embedding mapping and dictionary induction technique. Our experiments on bilingual lexicon induction and crosslingual word similarity show that our method is able to learn high quality bilingual embeddings from as little bilingual evidence as a 25 word dictionary or an automatically generated list of numerals, obtaining results that are competitive with state-of-the-art systems using much richer bilingual resources like larger dictionaries or parallel corpora. In spite of its simplicity, a more detailed analysis shows that our method is implicitly optimizing a meaningful objective function that is independent from any bilingual data which, with a better optimization method, might allow to learn bilingual word embeddings in a completely unsupervised manner.

In the future, we would like to delve deeper into this direction and fine-tune our method so it can reliably learn high quality bilingual word embeddings without any bilingual evidence at all. In addition to that, we would like to explore non-linear transformations (Lu et al., 2015) and alternative dictionary induction methods (Dinu et al., 2015). Finally, we would like to apply our model in the decipherment scenario (Dou et al., 2015).

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
