# Peer review of "Learning bilingual word embeddings with (almost) no bilingual data"

_ACL 2017 — decision unknown_

[Official Review · Reviewer 1 · rating 4 · confidence 4]
soundness 5 · originality 5 · clarity 4 · impact 3 · substance 4 · appropriateness 5 · meaningful comparison 3 · presentation format Oral Presentation

- Strengths:

The paper presents an iterative method to induce bilingual word embeddings
using large monolingual corpora starting with very few (or automatically
obtainable numeral) mappings between two languages. Compared to
state-of-the-art using larger bilingual dictionaries or parallel/comparable
corpora, the results obtained with the presented method that relies on very
little or no manually prepared input are exciting and impressive.

- Weaknesses:

I would have liked to see a discussion on the errors of the method, and
possibly a discussion on how the method could be adjusted to deal with them.

- General Discussion:

Does the frequency of the seeds in the monolingual corpora matter?

It would be interesting to see the partial (in the sense of after n number of
iterations) evolution of the mapping between words in the two languages for a
few words. 

What happens with different translations of the same word (like different
senses)?

One big difference between German and English is the prevalence of compounds in
German. What happens to these compounds? What are they mapped onto? Would a
preprocessing step of splitting the compounds help? (using maybe only
corpus-internal unigram information)

What would be the upper bound for such an approach? An analysis of errors --
e.g. words very far from their counterpart in the other language -- would be
very interesting. It would also be interesting to see a discussion of where
these errors come from, and if they could be addressed with the presented
approach.

[Official Review · Reviewer 2 · rating 4 · confidence 5]
soundness 5 · originality 5 · clarity 5 · impact 3 · substance 4 · appropriateness 5 · meaningful comparison 3 · presentation format Poster

This work proposes a self-learning bootstrapping approach to learning bilingual
word embeddings, which achieves competitive results in tasks of bilingual
lexicon induction and cross-lingual word similarity although it requires a
minimal amount of bilingual supervision: the method leads to competitive
performance even when the seed dictionary is extremely small (25 dictionary
items!) or is constructed without any language pair specific information (e.g.,
relying on numerals shared between languages). 

The paper is very well-written, admirably even so. I find this work 'eclectic'
in a sense that its original contribution is not a breakthrough finding (it is
more a 'short paper idea' in my opinion), but it connects the dots from prior
work drawing inspiration and modelling components from a variety of previous
papers on the subject, including the pre-embedding work on
self-learning/bootstrapping (which is not fully recognized in the current
version of the paper). I liked the paper in general, but there are few other
research questions that could/should have been pursued in this work. These,
along with only a partial recognition of related work and a lack of comparisons
with several other relevant baselines, are my main concern regarding this
paper, and they should be fixed in the updated version(s).

*Self-learning/bootstrapping of bilingual vector spaces: While this work is one
of the first to tackle this very limited setup for learning cross-lingual
embeddings (although not the first one, see Miceli Barone and more works
below), this is the first truly bootstrapping/self-learning approach to
learning cross-lingual embeddings. However, this idea of bootstrapping
bilingual vector spaces is not new at all (it is just reapplied to learning
embeddings), and there is a body of work which used exactly the same idea with
traditional 'count-based' bilingual vector spaces. I suggest the authors to
check the work of Peirsman and Pado (NAACL 2010) or Vulic and Moens (EMNLP
2013), and recognize the fact that their proposed bootstrapping approach is not
so novel in this domain. There is also related work of Ellen Riloff's group on
bootstrapping semantic lexicons in monolingual settings.

*Relation to Artetxe et al.: I might be missing something here, but it seems
that the proposed bootstrapping algorithm is in fact only an iterative approach
which repeatedly utilises the previously proposed model/formulation of Artetxe
et al. The only difference is the reparametrization (line 296-305). It is not
clear to me whether the bootstrapping approach draws its performance from this
reparametrization (and whether it would work with the previous
parametrization), or the performance is a product of both the algorithm and
this new parametrization. Perhaps a more explicit statement in the text is
needed to fully understand what is going on here.

*Comparison with prior work: Several very relevant papers have not been
mentioned nor discussed in the current version of the paper. For instance, the
recent work of Duong et al. (EMNLP 2016) on 'learning crosslingual word
embeddings without bilingual corpora' seems very related to this work (as the
basic word overlap between the two titles reveals!), and should be at least
discussed if not compared to. Another work which also relies on mappings with
seed lexicons and also partially analyzes the setting with only a few hundred
seed lexicon pairs is the work of Vulic and Korhonen (ACL 2016) 'on the role of
seed lexicons in learning bilingual word embeddings': these two papers might
also help the authors to provide more details for the future work section
(e.g., the selection of reliable translation pairs might boost the performance
further during the iterative process). Another very relevant work has appeared
only recently: Smith et al. (ICLR 2017) discuss 'offline bilingual word
vectors, orthogonal transformations and the inverted softmax'. This paper also
discusses learning bilingual embeddings in very limited settings (e.g., by
relying only on shared words and cognates between two languages in a pair). As
a side note, it would be interesting to report results obtained using only
shared words between the languages (such words definitely exist for all three
language pairs used in the experiments). This would also enable a direct
comparison with the work of Smith et al. (ICLR 2017) which rely on this setup.

*Seed dictionary size and bilingual lexicon induction: It seems that the
proposed algorithm (as discussed in Section 5) is almost invariant to the
starting seed lexicon, yielding very similar final BLI scores regardless of the
starting point. While a very intriguing finding per se, this also seems to
suggest an utter limitation of the current 'offline' approaches: they seem to
have hit the ceiling with the setup discussed in the paper; Vulic and Korhonen
(ACL 2016) showed that we cannot really improve the results by simply
collecting more seed lexicon pairs, and this work suggests that any number of
starting pairs (from 25 to 5k) is good enough to reach this near-optimal
performance, which is also very similar to the numbers reported by Dinu et al.
(arXiv 2015) or Lazaridou et al. (ACL 2015). I would like to see more
discussion on how to break this ceiling and further improve BLI results with
such 'offline' methods. Smith et al. (ICLR 2017) seem to report higher numbers
on the same dataset, so again it would be very interesting to link this work to
the work of Smith et al.

In other words, the authors state that in future work they plan to fine-tune
the method so that it can learn without any bilingual evidence. This is an
admirable 'philosophically-driven' feat, but from a more pragmatic point of
view, it seems more pragmatic to detect how we can go over the plateau/ceiling
which seems to be hit with these linear mapping approaches regardless of the
number of used seed lexicon pairs (Figure 2).

*Convergence criterion/training efficiency: The convergence criterion, although
crucial for the entire algorithm, both in terms of efficiency and efficacy, is
mentioned only as a side note, and it is not entirely clear how the whole
procedure terminates. I suspect that the authors use the vanishing variation in
crosslingual word similarity performance as the criterion to stop the
procedure, but that makes the method applicable only to languages which have a
cross-lingual word similarity dataset. I might be missing here given the
current description in the paper, but I do not fully understand how the
procedure stops for Finnish, given that there is no crosslingual word
similarity dataset for English-Finnish.

*Minor:
- There is a Finnish 'Web as a Corpus' (WaC) corpus (lines 414-416):
https://www.clarin.si/repository/xmlui/handle/11356/1074
- Since the authors claim that the method could work with a seed dictionary
containing only shared numerals, it would be very interesting to include an
additional language pair which does not share the alphabet (e.g.,
English-Russian, English-Bulgarian or even something more distant such as
Arabic and/or Hindi).

*After the response: I would like to thank the authors for investing their time
into their response which helped me clarify some doubts and points raised in my
initial review. I hope that they would indeed clarify these points in the final
version, if given the opportunity.

[Official Review · Reviewer 3 · rating 4 · confidence 3]
soundness 5 · originality 5 · clarity 5 · impact 3 · substance 4 · appropriateness 5 · meaningful comparison 3 · presentation format Oral Presentation

The paper presents a self-learning framework for learning of bilingual word
embeddings. The method uses two embeddings (in source and target languages) and
a seed lexicon. On each step of the mapping learning a new bilingual lexicon is
induced. Then the learning step is repeated using the new lexicon for learning
of new mapping. The process stops when a convergence criterion is met.

One of the strengths is that the seed lexicon is directly encoded in the
learning process as a binary matrix. Then the self-learning framework solves a
global optimization problem in which the seed lexicon is not explicitly
involved. Its role is to establish the initial mapping between the two
embeddings. This guarantees the convergence. The initial seed lexicon could be
quite small (25 correspondences).

The small size of the seed lexicon is appealing for mappings between languages
for which there are not large bilingual lexicons.

It will be good to evaluate the framework with respect to the quality of the
two word embeddings. If we have languages (or at least one of the languages)
with scarce language resources then the word embeddings for both languages
could differ in their structure and coverage. I think it could be simulated on
the basis of the available data via training the corresponding word embeddings
on different subcorpora for each language.